# The Hypothesis of the Prolonged Cell Cycle in Turner Syndrome

**DOI:** 10.3390/jdb10020016

**Published:** 2022-05-11

**Authors:** Francisco Álvarez-Nava, Marisol Soto-Quintana

**Affiliations:** 1Biological Sciences School, Faculty of Biological Sciences, Central University of Ecuador, Quito 170113, Ecuador; 2Genetic Research Institute, University of Zulia, Maracaibo 4001, Venezuela; mlizell@hotmail.com

**Keywords:** cell proliferation, congenital heart disease, embryonic lethality, folliculogenesis, neuropsychological profile, prolonged cell cycle, short stature, Turner syndrome

## Abstract

Turner syndrome (TS) is a chromosomal disorder that is caused by a missing or structurally abnormal second sex chromosome. Subjects with TS are at an increased risk of developing intrauterine growth retardation, low birth weight, short stature, congenital heart diseases, infertility, obesity, dyslipidemia, hypertension, insulin resistance, type 2 diabetes mellitus, metabolic syndrome, and cardiovascular diseases (stroke and myocardial infarction). The underlying pathogenetic mechanism of TS is unknown. The assumption that X chromosome-linked gene haploinsufficiency is associated with the TS phenotype is questioned since such genes have not been identified. Thus, other pathogenic mechanisms have been suggested to explain this phenotype. Morphogenesis encompasses a series of events that includes cell division, the production of migratory precursors and their progeny, differentiation, programmed cell death, and integration into organs and systems. The precise control of the growth and differentiation of cells is essential for normal development. The cell cycle frequency and the number of proliferating cells are essential in cell growth. 45,X cells have a failure to proliferate at a normal rate, leading to a decreased cell number in a given tissue during organogenesis. A convergence of data indicates an association between a prolonged cell cycle and the phenotypical features in Turner syndrome. This review aims to examine old and new findings concerning the relationship between a prolonged cell cycle and TS phenotype. These studies reveal a diversity of phenotypic features in TS that could be explained by reduced cell proliferation. The implications of this hypothesis for our understanding of the TS phenotype and its pathogenesis are discussed. It is not surprising that 45,X monosomy leads to cellular growth pathway dysregulation with profound deleterious effects on both embryonic and later stages of development. The prolonged cell cycle could represent the beginning of the pathogenesis of TS, leading to a series of phenotypic consequences in embryonic/fetal, neonatal, pediatric, adolescence, and adulthood life.

## 1. Introduction

The cell cycle is the period between successive cell divisions. This process encompasses a highly regulated series of events where DNA replication occurs (S phase) and each of the two resulting DNA copies are segregated properly during division cell (mitosis or meiosis; M phase). The other two critical events take place between the M and S phases: G1 following the M phase and G2 following the S phase. The orderly progression of the cell cycle is mainly mediated by transcriptional regulatory proteins.

Cell proliferation is the increment in the cell number due to mitotic cell division. Several factors influence cell proliferation, such as the initial number of stem cells, the number of cells that complete cell division, the total number of cell cycles, the cell cycle frequency, and the number of proliferating daughter cells that are produced. Homeostatically, diminished cell proliferation may be counterbalanced by an increase in cell volume, but during the critical embryonic period, the reduction in cell number could affect cell differentiation and organogenesis. Thus, the control of the speed and frequency of cell division is very important in tissues requiring a high rate of cell proliferation. For example, in the development of the central nervous system, the number of neurons generated during organogenesis is critical for the formation of nerve circuits in the brain [1]. Alterations in the proliferation of progenitor cells leading to a reduction in cell numbers and/or deficiencies in the differentiation of daughter cells could cause several congenital defects and/or conditions in postnatal life. Cell growth and differentiation are mechanisms that are complexly synchronized with developmental age and the metabolic, hormonal, and nutritional status in humans. In 45,X embryos, the development, size, and function of several organs and tissues could be severely affected by a decrease in the number of cells during the critical period of organogenesis due to aneuploidy.

A longer cell cycle time in 45,X cell lines than in 46,XX cells has been reported [2]. The doubling time [3] and cell generation time [4] parameters were shown to be prolonged in fibroblast lines with an absent or aberrant X chromosome compared to 46,XX cell lines. Simpson and Le Beau (1981) calculated the difference between the cell cycles of aneuploid cells (45,X), cell lines with structural X chromosome abnormalities, and eudiploid 46,XY and 46,XX cells on cultured skin fibroblasts. They found that the cell cycle of·45,X lines was significantly increased compared to that of their euploid controls. The mean values of the duration of the cell cycle (in hours) and its components (G1, S, and G2) were based upon four replicates initiated from each line (45,X (23.09 ± 0.92) and X-chromosome structural abnormalities (24.44 ± 1.62)) vs. 46,XX (18.19 ± 0.66) and 46,XY (17.93 ± 0.15)). The researchers reported that most of the increase was due to the prolongation of the S period [2]. Consequently, a longer intermitotic period in 45,X cells would lead to a deceleration of the cell growth rate. This suggests that 45,X cells have a failure to proliferate at a normal rate, leading to a decreased cell number in a given tissue during organogenesis. This reduced cell number is unable to sustain adequate cell differentiation, causing disturbances in developmental fields and leading to global developmental retardation [5]. The prolongation of the cell cycle in 45,X might predominantly affect the differentiation of tissues and organs at certain critical embryonic periods when cells must divide very rapidly, including embryonic stem cells from which the brain, heart, kidney, bone, blood vessels, Corti’s organ, pancreatic β-cells, and ovarian tissues are derived. Several clinical studies support this hypothesis in TS: (a) it is estimated that only 1% of 45,X embryos survive to term; (b) 45,X fetuses that survive experience intrauterine growth retardation (IUGR), and newborns are small for gestational age; and (c) 45,X subjects present short stature, several dysmorphic stigmata, and urinary, cardiovascular, skeletal and endocrine abnormalities. Thus, high embryonic lethality, IUGR, short stature, and other features seen in TS support the prolonged cell cycle hypothesis. A prolonged cell cycle with a reduced number of cells has also been observed in diseases of different etiologies, such as congenital rubella infection and Bloom syndrome, which present prenatal and postnatal growth retardation [6].

How aneuploidy or the absence of the second sex chromosome lengthens the cell cycle is a matter of speculation. Barlow suggested that 45,X may result in an “upset of the developmental programme” [7]. The complete or partial loss of the second sex chromo-some would alter several cell processes and the expression of hundreds of genes. Therefore, most of the phenotypic characteristics associated with monosomy X, including a prolonged cell cycle, could arise from the concurrent modification of various gene products (proteins, ncRNA) that could have a small individual effect. These individual effects may have cumulative actions that would produce a continuous effect on cell growth.

For example, in a mouse model, the single X chromosome in oocytes was shown to be asynapsized or “self-synapsized”. The asynapsed X chromosome oocytes were eliminated during diplonema. This elimination was associated with the presence of the phosphorylated form of the histone variant H2afx (γH2AFX) on the asynapsed X chromosome. The γH2AFX is an epigenetic mark involved in the transcriptional silencing of asynapsed chromosomes during meiosis. DNA double-strand break (DSB) foci disappeared on asynapsed chromosomes during pachynema, which rejected the DNA damage as a cause of germ cell loss at meiotic prophase I and demonstrate the existence in oocytes of a repair pathway for asynapsis-associated DNA DSB. Interestingly, mice carrying the H2afxS H2AX null allele transgene restored oocyte numbers in XO females to wild type XX levels. These changes may lead to retarded cell cycle and the epigenetic silencing of key X chromosomal genes whose expression is essential for the development/retention of oocytes such as Bmp15; Fmr1; Zfx [8,9].

On the other hand, the *HCFC1* (Host Cell Factor C1) gene encodes a nuclear coactivator involved in the control of the cell cycle and transcriptional regulation. *HCFC1* resides on the Xq28 region and is subject to X-chromosome inactivation. The role of *HCFC1* in cell proliferation is suggested due to its transcripts and protein being copiously expressed in fetal and placental tissues. The HCFC1 protein promotes the passage of the cell through the G1 phase and ensures proper cytokinesis in the M phase [10].

Thus, transcription factors that influence the cell cycle duration are present on the X chromosome, and haploinsufficiency may delay the cell cycle. Additionally, the haploinsufficiency of genes involved in DNA replication or repair on the X-chromosome would lengthen the cell cycle in 45,X cells, which could alter the capacity of these cells to respond to S replication signals.

If cell clones with 45,X, or X chromosomes with deletions have a prolonged cell cycle, a reduction in the number of cell clones per time is expected (Figure 1), leading to a selective disadvantage in vitro and probably also in vivo. In a serial culture with cocultivated 45,X and 46,XX cell lines, the proportion of 45,X cells was shown to be decreased, while the proportion of the 46,XX/46,XX cocultivated control lines was similar [4]. More recent evidence supports the idea that aneuploid cells are selected against/eliminated in vivo [11]. Aneuploid hematopoietic stem cells (HSCs) showed reduced fitness compared with euploid controls when transplanted into irradiated mice [12]. Additionally, the frequency of the 45,X cell line falls drastically after birth [13], although in the normal population, loss of one of the sex chromosomes leading to 45,X is a part of the aging process [14], this is the opposite of the scenario that represents 45,X/46,XX mosaicism in TS. In a 10-year longitudinal study, the proportion of diploid 46,XX cells increased with time in women with TS, and the percentage of increase in diploid cell proportion was positively correlated with age [15,16]. Thus, 46,XX cell clones probably divide faster than 45,X cells. Consequently, tissues with high cell turnover would show decreased function as the age of the individual increases owing to the decrease in the number of cells. This is due to a prolonged cell cycle and/or the selective disadvantage of 45,X cell clones. Adult stem cells that form tissues and organs such as bone, gonads, Corti’s organ, and pancreatic β-cells would be especially susceptible. In the following paragraphs, lines of evidence associating the clinical manifestations in adult TS individuals with the hypothesis of the prolonged cell cycle will be described.

## 2. XO Monosomy in Animals

Monosomy X has been reported in several animal species, including horses, rhesus monkeys, cows, buffalo, sheep, dogs, cats, alpacas, and pigs [17,18]. No experimental information is available in XO animal models about the behavior of the cell cycle in aneuploid cells. However, several phenotypic features are common in XO eutherians mammals with 45,X subjects, including a short stature (short body and legs), small gonads lacking follicular development, and irregular or absent estrus cycle. All XO mammal animals studied so far were described as sterile. Thus, if there were a gene or loci within the sex chromosomes that explained the TS phenotype, this gene(s) would be present in all the species listed above. Eutherian mammal and human species share evolutionarily different classes of genes on the sex chromosomes that are expected to largely affect various tissues, organs, and systems during embryonic development, growth, and adult life and are likely to cause TS phenotypes under the Gene Dosage Effect hypothesis. This gene(s) must also fulfill certain criteria to be considered as a contributor(s) to the TS phenotype, including to be transmitted of pseudoautosomal manner where the Y-chromosome gene(s) is functionally equivalent to the gene on the X-chromosome; to escape X-chromosome inactivation, and to have a ubiquitous expression. The best location for this gene(s) on the genome would be the pseudoautosomal regions (PARs), which are limited regions of identical sequence on the mammalian sex chromosomes with known functions in chromosome pairing, recombination, and segregation in meiosis of the heterogametic sex [19]. Mapping analyses indicate that the PARs vary in size and gene content between eutherian species (alpaca, cat, cattle, dog, horse, pig, and rabbit) [17]. These differences might critically influence the genetic effects in the TS phenotype. However, the phenotype caused by the loss of the second sex chromosome among the different mammal species is almost constant. Differences observed between the phenotypes among animal and human species may be attributed to genes that are known to be on the human sex chromosomes that are not present on that of the other species. In humans, ~15% of X-linked genes escape X-inactivation [20,21], so only a small number of genes are predicted to contribute to dosage imbalances in 45,X monosomy. However, the only gene that has been proven to be associated with clinical features of TS (skeletal anomalies and short stature) is the SHOX gene (short stature homeobox-containing gene, NM000451) [22]. Thus, the variability in size and gene content in PARs among species of eutherian mammals may contradict the deterministic hypothesis of a TS gene(s) and advantage a more general mechanism such as a prolonged cell cycle. The latter could explain the common phenotypic similarities among animals with XO monosomy despite genetic differences.

Unlike, XO mice do not display many of the phenotypic characteristics common to X chromosome monosomy in human and other eutherian mammalian species. Also, XO mice have no major congenital defects [23] and are fertile. Therefore, several authors argue that XO mice cannot be used as a model to study the TS phenotype [24]. However, the XO mice are a useful model of TS because it is the only genetically tractable mammalian species in which viable female individuals with one or two X chromosomes can easily be produced [25]. For example, the XO mice have fewer oocytes compared to normal XX mice and exhibit premature ovarian failure [25]. Also, evidence indicates that maternal monosomy X has deleterious effects on preimplantation embryo development [26]. Similarly, the growth and development of XO mice are normal, although this contrasts with their early prenatal growth. XO embryos that retain the paternal X chromosome are developmentally delayed early in pregnancy [26]. Remarkably, a distinctive neurocognitive phenotype is described in XO mouse models similar to that described in 45,X subjects [27].

## 3. Clinical Consequences of Reduced Cell Proliferation of 45,X Cells

### 3.1. Embryonic Lethality

The loss of one of the two sex chromosomes due to nondisjunction or anaphase lag during gametogenesis or early embryogenesis is a relatively common event that contributes to the formation of zygotes or embryos monosomic for sex chromosomes [28]. The monosomy of the second sex chromosome is considered the most frequent genetic abnormality in the human species, as it is present in approximately 2% of all conceptions that survive long enough to be clinically recognized pregnancies [29]. However, there is a well-described increase in intrauterine lethality. Approximately 9–15% of spontaneous abortions and 0.25% of stillbirths are 45,X [30]. Additionally, the incidence of 45,X pregnancies over the course of pregnancy is highly variable, with prevalence rates of 6.7, 3.2, and 2.17 per 10,000 female fetuses at 12, 20, and 40 weeks gestation, respectively [31]. Thus, the probability that a 45,X gestation survives to term is just 0.3%. The high lethality of 45,X conceptuses occurs especially during the implantation period [30], which suggests the involvement of rather general etiopathogenic mechanisms [32]. Indeed, slow-growing blastocysts have lower implantation potential when transferred in fresh cycles [33]. Most 45,X abortions in the mid to late first trimester present with ruptured chorionic and amniotic sacs, and minimal or very delayed fetal development [34].

The embryonic and fetal survival of 0.01% of the 45,X concepts can be explained by the acquisition of monosomy of the sex chromosomes. This is a relatively common event in early embryonic development. Chromosomal instability is a well-known feature in blastomeres. This chromosomal instability involves monosomy–trisomy for whole chromosomes and other genomic and chromosomal mutations [35]. It has been proposed that a large proportion of 45,X embryos postzygotically acquire their 45,X cell line [36]. Although cryptic mosaicism cannot be detected in conventional cytogenetic analysis from peripheral blood lymphocytes and other tissues, a placental rescue line could explain embryonic and fetal survival. Small (placental) marker sexual chromosome rescuing has also been proposed to elucidate this survival [30,37] In addition, one hypothesis to explicate the survival of the postzygotic 45,X cell line in contrast to the normal euploid cell line (46,XX or 46,XY) is that epigenomic deregulation could promote rapid survival selection of 45,X cells. Thus, the loss of a second sexual chromosome may introduce a global epigenetic effect as a cellular response to overcome the detrimental effects of monosomic cells [38]. This may indirectly facilitate the process of adaptive cells by producing a fitness advantage only after a secondary and required genetic event has occurred [30].

Three nonexclusive hypotheses may explain the high frequency of early lethality in 45,X embryos. First, in both the morula and blastocyst stages, the pluripotent 45,X cells are unstable because cell proliferation is triggered very slowly. In this scenario, the 45,X cells would not transform into embryonic stem cells and would automatically degenerate. At a more advanced embryonic stage, the second hypothesis postulates that there is a differentiation blockage of 45,X cells and, consequently, certain tissues and/or structures cannot differentiate properly, leading to embryonic loss, or when compatible with implantation, it can also cause serious fetal complications, such as IUGR and birth defects. A third scenario seems to be more feasible: inadequate global placentation due to disturbed cell proliferation/differentiation may disrupt coordinated embryonic and extraembryonic tissue growth, eventually causing embryo death. Thus, following the intrinsic delay of embryonic cellular growth, the inability to proliferate at a normal rate could explain the high embryonic lethality of 45,X conceptuses [2]. In addition, the presence of a smaller number of cells in a developing organ can cause worsening in cell differentiation and organogenesis, leading to failure in implantation and placentation as well as birth defects and, in turn, resulting in embryonic death and spontaneous abortion.

Although 45,X cells can take part in normal prenatal and postnatal development, they can fail at certain critical stages where very rapid cell growth of a small group of stem cells is required. These periods include early (cleavage, blastogenesis, implantation, placentation) and late embryonic (gastrulation, neurogenesis, somitogenesis, and organogenesis) and early postnatal development. These developmental windows are critical times for neural, skeletal, mesodermal, and mesenchymal histogenesis. Therefore, the prolonged cell cycle could explain the placental insufficiency, IUGR, short stature, and dysmorphogenesis seen in TS.

### 3.2. Growth Retardation, Short Stature, and Osteopenia/Osteoporosis

Intrauterine growth retardation (IUGR) has been extensively documented in TS, and this growth deficiency becomes evident by the middle of the second trimester of gestation [39]. However, it is not fully understood when IUGR begins and what its causes are in TS. Despite this fetal growth deficiency, most newborns with TS regularly present with normal birth weight and length for gestational age. The data published so far agree that the growth deficit in most newborns with TS is approximately −1,0 SDS [40,41,42] and the deviation from normal is relatively small. Similarly, XO mouse embryos develop more slowly than XX embryos until early mid-gestation, but they reach the same stage in their growth and development as their XX littermates at birth [25]. Ishikawa et al. reported the mean number of somites was greater in XX mouse embryos than it was in XO mouse embryos [43]. In addition, XO mouse embryos have placentae that are larger than XX placentae in late pregnancy [44], although in early pregnancy the ectoplacental cone, a placental precursor, is smaller in XO mice than in their XX sibs [45]. This finding suggests that enlargement of XO placentae in late gestation could be a consequence of a compensatory response to initial ectoplacental cone deficiency [46]. The essential effect of monosomy of sex chromosomes could be in extending DNA replication time [47]. To illustrate, the trophoblast giant cells uniquely undergo multiple rounds of the S period without cell division [48]. This can have a significant impact on placental growth. If a prolonged S period gives rise to reduced ploidy in the trophoblast giant cells, then a reduction in gene expression in these cells could result, with a consequent overall reduction in placental growth [47]. It is striking that in the TS animal model, XO mouse embryos have a delay in the somitogenesis and early development of the placenta. These are simultaneous events that occur in a crucial period of embryonic growth where the rate of cell proliferation is critical. It is unknown if such events occur in the growth of early 45,X embryos and if compensatory mechanisms to prevent progressive deterioration in growth are triggered as occurs in XO mouse embryos. On the other hand, human fetal growth may be considered a multifactorial quantitative trait. Thus, IUGR in TS may be associated with many diseases and conditions (hydrops fetalis, placental insufficiency, congenital heart disease, renal malformations, etc.) that may occur at any time of fetal development. Also, polygenic factors contributing to this complex trait may be present in TS. The SHOX protein is believed to be associated with bone growth and development in fetal life as this gene is expressed in extremity buds as early as 32 days postconception [49]. Nevertheless, convincing evidence of fetal growth disorders in TS result from haploinsufficiency of the SHOX gene is lacking [50] and no information about this gene and its effects on fetal growth has been reported. In this aspect, subjects with 46,X,del(Xp) karyotypes and normal anthropometric parameters at birth were reported, and subsequently, they developed short stature [51,52]. Pseudoautosomal region 1 (PAR1), where the SHOX gene resides, is on the Xp chromosome. These patients have SHOX gene haploinsufficiency (but not 45,X), and were born with normal anthropometric variables. Therefore, other factors involved in controlling fetal growth may explain the difference in growth retardation before and after birth, including genes located either in the centromere or in the long arm of the X-chromosome. Also, evidence that postnatal growth retardation in XO mice is due to haploinsufficiency for a non-PAR X gene [53].

Alternatively, the fetal growth in TS phenotype could be the result of global chromosomal imbalance, rather than the addition of the effects of individual loci [54,55]. This chromosomal imbalance can not only affect the cell proliferation rate in critical periods of organogenesis such as somitogenesis but also the 45,X cells may lead to a number of epigenetic modifications in the genes controlling the growth intrauterine. Transcriptome analyses in different human tissues and XO mouse models did not reveal a direct correlation between genomic imbalance and gene expression levels [56,57,58,59,60]. Changes in gene expression have also been detected in loci that are not on the X chromosome and diverge between cell and tissue types [56,57,58,59,61]. Sharma et al. reported a downregulated gene expression of IGF2 (Insulin-like Growth Factor 2), in adult somatic cells with 45,X [60]. IGF2, a gene located on chromosome 11p15.5, is believed to be a major fetal growth gene in mammals in contrast to IGF1, which is a major postnatal growth factor. IGF2 is an imprinted gene, expressed only from the paternal allele. Maternally imprinted genes that are paternally expressed are supposed to promote growth either in utero or in the perinatal period. IGF2 was found to be upregulated in IUGR and spontaneous miscarriages cases at the second trimester [62]. This would mean that the “parental conflict” theory could not be applied to all imprinted human genes or all situations, namely, in IUGR or aneuploidy gestations. The status of gene expression IGF2 in 45,X fetuses is unknown, but an epigenetic compensatory mechanism could be triggered to inhibit the silencing of imprinted maternal genes to promote fetal growth in the second semester of human gestation.

Bone development and growth involve bone-forming cells (osteoblasts) and bone-resorbing cells (osteoclasts) acting on the surface of the bone or cartilage (chondrocytes) and embedded cells (osteocytes) within the bone matrix. All four cell types operate in a coordinated manner within the bone microenvironment (Figure 2). Osteogenesis comprises proliferation, extracellular matrix development and maturation, mineralization, and apoptosis. This process is primarily mediated by osteoblasts. Both the proliferation and differentiation of osteoblast progenitors are essential for bone growth and development. During normal bone formation, an increased rate of osteoblast proliferation and differentiation is necessary. Thus, bone mass depends on both the number and activity of osteoblasts. The osteoblast number is determined by the replication rate of mesenchymal stem cells and their lifespan, which is established by the timing of death by apoptosis [63].

#### 3.2.1. Short Stature

Short stature is the most common clinical feature of TS and may occur in the absence of other clinical manifestations [64], causing reduced bone mass. Several distinct phases of abnormal growth have been described in subjects with TS [42]. Growth failure begins in utero with mild IUGR, followed by a slow early childhood growth rate, which is more evident in late childhood and adolescence due to a delayed onset of the childhood phase of growth and the absence of a pubertal growth spurt in subjects with TS without pubertal development. This leads to a significantly severe short stature in adulthood with a final height of approximately 20 cm below the mean normal adult female height for the healthy population [65]. The most critical phase of height deficit occurs during the first 3 years of life [42].

Although the etiology of short stature is not well understood, growth impairment seems to be due to multiple factors. Haploinsufficiency (loss-of-function mutation of one allele) of the *SH*ort stature *H*omeob*OX*-containing gene (*SHOX*) is expected to be the most significant contributing factor to the short stature [22]. However, such an assumption is inaccurate because it does not fully explicate the great clinical variability in the growth pattern among individuals with TS. Thus, other factors, such as abnormalities in the GH-IGF-I axis, loss of quantitative trait loci in Xp22.3, estrogen deficiency, and general chromosomal imbalance, might be involved in TS-associated short stature [66].

The interindividual variability of growth impairment in TS is similar to the growth variance in the non-TS population. Additionally, the phases of growth impairment are almost constant among people with TS. Such findings cannot be explained by loss-of-function mutations of the *SHOX* allele. As an alternative, the interindividual variance of height is best explained by the presence of more general aneuploidy effects. Increasing clinical evidence suggests that growth-related phenotypes in TS are mediated by the effects of aneuploidy on cell turnover rates [67]. First, individual growth is independent of karyotype/phenotype [54]. Second, ethnic influences on the growth of girls with TS in different regions are reported [65]. Third, parental height correlates well with the final adult height, as previously observed in the normal population [68,69]. This suggests that the genetic height background (stature polygenes) is preserved in TS, despite continual growth retardation. Fourth, interindividual variation in short stature with or without a spectrum of skeletal anomalies is seen in subjects with TS [54]. Last, the degree of growth impairment noted in Leri-Weill syndrome (loss-of-function mutation of one *SHOX* allele) is not as severe as that described in TS [70].

Due to the general chromosomal imbalance, the cell cycle is longer in an aneuploid cell than in a euploid cell, leading to growth retardation, and this is more evident if the missing chromosome includes transacting growth-regulating genes such as *SHOX*. Analysis of the gene expression revealed *SHOX* expression in multiple fetal tissues (bone, skeletal muscle, heart, brain, bone marrow fibroblasts). It is precisely at the phases in which growth is affected in TS that the cell cycle could fail to be further upregulated to provide enough growth-supporting cells due to a delay in the cell cycle.

#### 3.2.2. Osteopenia/Osteoporosis

The pathogenesis of the reduced bone mass in TS remains unclear. Several mechanisms likely contribute to its ontogeny, including the disorder per se, bone dysmorphogenesis, haploinsufficiency of X chromosome-linked genes (e.g., the *SHOX* gene), abnormalities in the production, secretion, and/or action of hormones, and low physical activity. Thus, two mechanisms seem to be present in the reduction of bone mass seen in TS, an intrinsic factor that occurs early (pre-or postnatally) leading to a decrease in bone size and alterations in hormonal factors that develop later. Abnormalities in the GH–IGF–I axis in pubertal girls with TS have been reported [71,72,73]. Additionally, subjects with TS are deficient in androgens [74]. Furthermore, since estrogens increase bone accretion during normal puberty, the absence of rising estrogen levels or the delay in starting estrogen treatment plays a major role in the reduced bone mass in most pubertal and postpubertal individuals with TS. However, although several hormonal factors (the GH–IGF–I axis, gonadal steroids, calcium-regulating hormones) appear to be involved, the clinical, radiological, and biochemical data support that an intrinsic bone defect appears to be the main factor in the pathogenesis of reduced bone mass (osteopenia/osteoporosis) in TS [75]. Reduced markers of bone formation associated with a deficit in cortical bone are found early in childhood in TS individuals, which favors the idea of an intrinsic bone defect. Additionally, a decrease in trabecular bone volume and lack of active bone-forming surfaces were described in biopsy specimens from prepubertal girls with TS. Furthermore, a coarse trabecular pattern of the carpal bones, deficient periosteal growth, a failure to shift to endosteal bone formation, and osteopenia detected by dual-energy X-ray absorptiometry (DEXA) and quantitative computer-assisted tomography (QCT) were described in prepubertal patients with TS [76,77]. Thus, the data of markers of bone metabolism, histology, DEXA, and QCT in non-estrogen-treated TS subjects seem to suggest suboptimal bone accretion due to an intrinsic factor.

Several fracture risk factors in patients with TS appear to be present, including bone fragility (regardless of bone density), small bones, altered bone geometry, falls and traumatic episodes, impaired vestibular function, compromised balance, and reduced vision [78]. Although osteoporosis and fractures did not differ between women with the 45,X karyotype and those with other karyotypes [79], a higher and close to normal spine bone mineral density (BMD) were found in patients with TS with 45,X/46,XX compared to individuals with TS with the 45,X karyotype [80]. A significant deficit in cortical radial bone mineral density is seen in prepubertal TS subjects when compared with age- and body mass index (BMI)-matched healthy subjects, but this radiological finding disappears by adjusting for height [74]. Despite a normal volumetric BMD, a low areal BMD due to small bone size may lead to increased bone fragility in TS [81]. The small bone size was found to be a risk factor for fractures in postmenopausal women [82]. Thus, an association between height (or at least small bone size) and reduced bone mass (osteopenia/osteoporosis) is present in subjects with TS. Consequently, a similar mechanism could be responsible for short stature and osteopenia/osteoporosis.

Osteoblasts can have three fates; remaining on the bone surface as bone lining cells, differentiating into osteocytes, or undergoing apoptosis, but apoptosis is the main fate of osteoblasts [83] (Figure 2). Approximately 60% to 70% of osteoblasts die by apoptosis. When osteoprogenitor cells (preosteoblasts) are differentiated into osteoblasts, they immediately secrete the organic component of bone matrix and noncollagenous proteins (osteoid) and regulate their mineralization. Osteoblasts produce alkaline phosphatase, which is involved in bone mineralization and is an early marker of osteoblast differentiation. Additionally, osteoblasts express receptors for growth factors on their cell membrane. Ligand-receptor binding promotes the proliferation of osteoblasts to increase the bone matrix. Reduced osteogenesis or mineralization, regardless of whether it is due to a reduction in osteoblast number, osteoblast enzyme activity, or increased osteoblast apoptosis, leads to impaired bone mass. Therefore, an intrinsic bone defect in TS can lead to both reduced proliferation and an increased apoptosis rate of osteoblasts. Reduced osteoblastogenesis secondary to a cell cycle delay would give rise to decreased osteoblast recruitment of enough magnitude to explain the decline in bone mass in TS. Since the prolonged cell cycle of 45,X cells results in fewer cell clones per time, the proliferating osteoblast progenitor population is affected, leading to a reduction in the number of matrix-secreting osteoblasts. Decreased proliferation of preosteoblasts might affect not only bone mineralization but also skeletal growth and development, thus leading to a significant reduction in bone size with a disproportionate geometry (bone dysmorphogenesis and short stature) [84,85]. According to the prolonged cell cycle hypothesis, the population of osteocytes would be less compromised than that of osteoblasts since they do not undergo cell proliferation and may stay in G0 of the cell cycle for decades in the bone matrix. However, because all osteocytes are derived from osteoblasts, the osteocyte population is also compromised (Figure 2). It is unknown whether 45,X osteoblast apoptosis is increased, which has been reported in other tissues, but both less osteoblast proliferation and increased apoptosis probably occur in the bone microenvironment of 45,X cells.

### 3.3. Congenital Heart Disease

Subjects with TS have a substantially higher risk (50%) of congenital heart disease (CHD) than the general population [64], involving mainly the chambers and vessels of the left side of the heart. Cardiovascular disease is the principal causative factor of decreased life expectancy in TS [86]. In addition, congenital cardiovascular defects are responsible for fetal death by the second trimester in TS [87].

A significant association between the presence of *pterygium colli* and cardiovascular defects in TS has been reported in several studies [88,89,90]. Cystic hygromas are malformations commonly detected in fetuses with the 45,X karyotype. These multiloculated flu-id-filled sacs are thought to arise from a failure of the lymphatic system to communicate with the venous system in the fetal neck, often progressing to hydrops fetalis. However, this may resolve in utero, leading to a “jugular lymphatic obstruction sequence” in fetuses that survive [91]. Likewise, when lymphatic obstruction expands into the mediastinum, it distends the thoracic ducts, and the increased hydrostatic pressure would compress the developing aortic arch and disrupt intracardiac hemodynamics, increasing the resistance to left-sided blood flow in great vessels. Redirection of intracardiac blood flow would lead to flow-related congenital heart diseases such as bicuspid aortic valve, coarctation of the aorta, hypoplastic left heart, patent ductus arteriosus, aortic atresia or hypoplasia, aortic valve anomalies, persistent left superior vena cava, and anomalous pulmonary venous return [88]. This hypothesis is supported by several studies [88,90,92,93,94]. Although this association arose from retrospective studies and may reflect the most severe phenotype in 45,X individuals, suggesting a likely bias, a prospective study with a large cohort of subjects with TS not selected for cardiovascular disease confirmed this significant association independent of karyotype and parental imprinting [90].

Although mechanical factors may slow lymph vessel development [95], the evidence seems to indicate that the cascade of events leading to jugular lymphatic obstruction sequence in TS is determined by a global factor due to a retardation in the development of the early lymphatic primordia. Hypoplasia of the lymphatic vessels is probably a basic disorder in TS. The histopathological pattern is described as lymphatic vessels that are wide and thin-walled, possessing few or no valves [95,96]. The etiology of hypoplastic lymph vessels may be intrinsic to lymphatic endothelial cells and related to cell proliferation. It seems more probable that lymphatic obstruction in the early development of a 45,X conceptus may cause impairment of lymph flow and lymphedema, leading to increased nuchal translucency, cystic hygroma, fetal hydrops, and fetal death. Regardless of whether it is an intrinsic or extrinsic causative factor, the outcome is the same: extensive inhibition of lymphatic development, resulting in hypoplasia and localized agenesis of lymph vessels. The main factor that directs the proliferation and migration of lymphatic endothelial cells is vascular endothelial growth factor C (VEGF-C) during embryogenesis.

Nevertheless, *pterygium colli* and CDH do not always coexist [89,90,97]. Additionally, dilated vessels were demonstrated not only around the ascending aorta [92] but also around the pulmonary trunk [96]. Taken together, these findings suggest that other major causes of CHD should also be considered. Alterations in the migration of neural crest cells and cellular processes, including growth, differentiation, apoptosis, and extracellular matrix determination, may be involved in the pathogenesis of CHD.

Although altered blood flow was expected to disturb aortic arch development, heart and pharyngeal arch vessel morphogenesis were altered after ablation of the cranial neural crest before outflow tract septation occurred [97]. The cranial neural crest generates ectomesenchyme, which colonizes pharyngeal arches III, IV, and VI (Figure 3). Thus, the growth, migration, and differentiation of neural crest cells may play a key role in the development of CHD in TS since these cells differentiate into smooth muscle cells in the aortic arch, and the semilunar valve seems to be the destination of their migration through the pharyngeal arch. A hypothesis where 45,X/46,XX mosaic hemangioblast derivatives, a common mesodermal precursor, secrete a paracrine developmental factor to promote aortic valve remodeling is postulated by Prashark [98]. This hypothesis assumes relatively few cells may need to function normally for valve development to proceed. Thus, decreased proliferation cellular due to prolongated cell cycle would not explain the high prevalence of left-sided congenital heart defects in TS. However, in this model, the relationship between percent mosaicism and valve defects may be unpredictable and subject to stochastic variation. Therefore, adequate cell proliferation of neural crest cells that invade the endocardial cushions may also be required for the development of the aortic valve [99].

Consequently, alterations in the cellular process of neural crest cells would explain the coexistence of anomalies in the structures derived from these arches, such as coarctation of the aorta, bicuspid aortic valve, and hypoplasia of the left ventricular outflow tract in TS. Likewise, considering this hypothesis, the presence of a small number of smooth muscle cells in the tunica media of the hypoplastic aortic arch of fetuses with TS would be understood [94]. In addition, a dissecting aortic aneurysm would be an expression of the fragility of the aortic wall due to the decreased tunica media, which is entirely derived from cells of the cranial neural crest. Similarly, the association between dissecting aortic aneurysm and the bicuspid aortic valve is frequently reported in TS [94]. Furthermore, as neural crest cells can differentiate into several cell types, giving rise to distinct tissue structures, a proliferative defect before their migration could have more extensive effects than alterations that occur after their departure from the neural tube. Consequently, the decreased proliferation of 45,X cells may cause an increased association between CHD and the head/neck region anomalies observed in TS. Either a proliferative defect or changes in intracardiac blood flow resulting from impaired migration of the neural crest cells, name-ly, a reduced number or delayed differentiation of these cells, would cause defects in the chambers and vessels of the left side of the heart.

Abnormalities in the proliferation, migration, or differentiation of neural crest cells also appear to be related to facial and skin clinical features and abnormal development of the parathyroid gland and thymus in subjects with TS or 45,X fetuses. Abnormal development of the parathyroid gland and hypoplastic thymus in 45,X fetuses were described [94], which also suggests hypoplastic development of the IV pharyngeal pouch and arch.

As neural crest cells migrate and differentiate into various cell types, they are embedded in a cell-free matrix. These cells sustain the extracellular matrix through a complex interaction of secretory and proliferative signals [100]. The cellular processes (proliferation, migration, and differentiation) may be hindered by abnormal extracellular matrices. Anomalous lymph vessels, hypoplastic thymus, and CHD were reported to be related to the irregular migration of neural crest cells and abnormal distribution of extracellular matrix in mouse trisomy 16 [101]. Aberrant interactions between neural crest cells and a reduced extracellular matrix may also be present in 45,X embryos [94]. On the other hand, Recently, Corbitt et al. performed whole-exome sequencing on 188 TS subjects, and they found gene variants on TIMP3, an autosomal gene located at 22q12.3 was associated with bicuspid aortic valve and increased aortic dimensions in TS individuals [102]. These authors also evaluated Xp genes and observed that hemizygosity for TIMP1, a functionally redundant paralogue of TIMP3 located on Xp11.3, increased the odds of having a bicuspid aortic valve compared to heterozygous individuals. Thus, the synergistic combinatorial effect of a TIMP1 hemizygosity and TIMP3 risk alleles increased the risk for the bicuspid aortic valve to nearly 13-fold. Both TIMP1 and TIMP3 peptides are natural inhibitors of the matrix metalloproteinases, a group of peptidases involved in the degradation of the extracellular matrix, and they also promote cell proliferation in a wide range of cell types and may also have an anti-apoptotic function. Similarly, SMADs are a family of structurally similar proteins that function as second messengers by transferring the signals from specific receptors to the target genes of the transforming growth factor-β (TGFβ) system. Thus, this axis works in a coordinated fashion to direct cellular processes (cell growth, migration, differentiation, and death) during embryonic development and recurrently throughout life in response to extrinsic signals. The target genes of the TGFβ system include genes of matrix proteins, collagen, fibronectin, and matrix metalloproteinases [103]. Several studies depicted a key role of the TGFβ system in the pathogenesis of aortopathy in many disorders, regardless of etiology [104]. All pharyngeal IV arch arteries derive TGFβ2-expressing smooth muscle cells from the neural crest [103]. Thus, the increased vulnerability of the IV arch arteries in TS can be associated with a small number of neural crest cells and related to the expression of fibronectin and neural cell adhesion molecules. A profound perturbation of the TGFβ system was reported in the circulation of individuals with TS [105]. Additionally, although low blood flow and pressure may adversely influence endothelial cells, reduced TGF-1 secretion is possible in fetuses with TS due to a lower number of smooth muscle cells in the tunica media.

### 3.4. Gonadal Dysgenesis

Although spontaneous puberty occurs in 15–30% of girls with TS [106], only 5% of them experience menarche [107], and 2% of women with TS can conceive [108,109,110]. In most women with TS, accelerated oocyte atresia leads to severely impaired folliculogenesis and, consequently, ovarian failure before puberty. Usually, these women with TS have a non-mosaic 45,X karyotype, as monosomic germ cells are unable to complete meiosis and are consequently eliminated during germ cell development [111,112]. Thus, ovarian function is probably dependent on the presence of 46,XX cells [108,113]. Ovarian follicles develop, not from 45,X germ cells but from 46,XX germ cells [36,114]. Most likely, women with TS who present with menarche have either a mosaic peripheral blood karyotype 45,X/46,XX or hidden somatic mosaicism in the gonadal tissue [115]. In approximately 30% of women with TS, the peripheral blood karyotype is a mosaic cell [116], where a 45,X cell line coincides with one non-45,X cell line. The 45,X/46,XX karyotype is the most frequent mosaic type, but other forms of mosaicism may be present. Somatic mosaicism is a major modifier of TS features and may be more prevalent than once thought [117]. Mosaicism varies with tissue type and patient age [6] and is associated with a milder degree of ovarian dysfunction [117]. A body of evidence supports this assumption. First, a higher prevalence of spontaneous menarche was observed in 45,X/46,XX than in 45,X patients [107]. Second, a greater follicular count was found in individuals with the lowest degree of mosaicism among girls with TS [118]. A distinct difference in the patterns of FSH secretion between TS girls with 45,X and 45,X/46,XX was also demonstrated [119]. Therefore, functional ovarian tissue in women with TS will depend on the presence of 46,XX germ cells in their ovaries.

Follicular ovarian development begins when an increasing number of oogonia enter the leptotene stage of meiotic prophase and become enclosed by a layer of somatic cells. Oogonia undergo rapid proliferation to reach 6 × 10^6^ by mid-trimester, but a significant drop follows, and they are reduced to 2 × 10^6^ by the term [115,120]. Thus, during normal gonadal development, germ cells undergo proliferation and apoptosis. The differentiation and migration of 45,X germ cells normally take place in early genital crests, but a significant decrease in germ cell numbers occurs by mid-gestation, leading to streak gonads at birth [118]. Apoptosis in 45,X germ cells is increased. TUNEL analysis showed that apoptosis was present in 3–7% and 50–70% of germ cells of 20-week-old normal and TS fetuses, respectively [111]. Rare oogonia but no evidence of primordial follicle formation was found in 45,X fetal ovaries at 25–37 weeks of gestation [121]. Thus, a massive oocyte decline occurs in the gonads of 45,X fetuses during fetal life.

The reduced 45,X germ cell number may be due to decreased mitotic proliferation or defective meiotic pairing [120]. It is unclear whether an increased frequency of meiotic pairing errors may cause 45,X germ cell apoptosis, as arrest in oogenesis occurs before chromosome pairing is established [122]. Additionally, it has been shown that in XO mice, the ovaries are anatomically normal, but although fertile, they suffer from premature ovarian failure [21]. Another explanation for impaired folliculogenesis is the inability to obtain normal follicle assembly (Figure 4). The defective proliferation of granulosa cells might alter the coupling of granulosa cells among themselves and with the oocyte, which might lead to germ cell apoptosis [123]. Among the functions of granulosa cells are the production of sex steroids and other growth factors stimulating oocyte development. In a study that evaluated whether ovarian tissue cryopreservation is a realistic option to preserve fertility in subjects with TS, 42 of the 46 oocytes (91%) analyzed in the small ovarian follicles detected in TS patients had a normal X chromosomal content (46,XX), but the granulosa cells were largely 45,X [124]. Consequently, although the oocytes from patients with TS could be 46,XX, this does not ensure that these follicles are functional, as the 45,X granulosa cells would not control the meiotic arrest of the oocyte and/or would not support normal follicular maturation. At least 10 mitotic divisions are needed to produce enough granulosa cells to form a mature antral follicle [125]. Thus, due to a prolonged cell cycle, the 45,X granulosa cell clones may fail to expand at a crucial time where cell proliferation is required (Figure 4).

In summary, the causes of the decline in germ cell number, leading to ovarian failure in TS, can be explained through the prolonged cell cycle hypothesis: (a) decreased mitotic proliferation of 45,X germ cells; (b) delay and degeneration of meiotic progression; and (c) reduced proliferation of granulosa cells, which disrupts the formation of ovarian follicles.

### 3.5. Impaired Pancreatic β-Cell Function

The pancreas is embryologically and functionally a dual organ. Functionally, the pancreas is both an endocrine (secretes insulin and other hormones) and an exocrine (digestive) organ. Embryologically, the pancreas develops from the fusion of two diverticula of primordial gut tissue (ventral and dorsal endoderm). The pluripotent pancreatic stem cells successively differentiate into committed multipotent pancreatic stem cells for the endocrine and exocrine lineages [126]. Thus, the embryological development of the pancreas is regulated by a complicated interaction of transcription factors and signaling pathways that establish the differentiation of exocrine and endocrine lineages. Pancreatic human embryonic stem cells differentiate into five types of hormone-secreting cells, of which α (glucagon) and β (insulin) glucose-responsive cells are the most important cells since their respective hormones regulate blood glucose. A common pancreatic stem cell independently gives rise to the α- and β-cell lineages. Differentiation of this multipotent stem cell towards β-cells is carried out in a step-by-step manner [127].

During the embryonic and fetal period, the cellular growth, differentiation, and specialization of numerous tissues, including pancreatic β-cells, are dependent on the cell replication rate (Figure 1). The greatest cell mass volume of pancreatic β-cells is possibly reached during the fetal and early postnatal periods, a critical but brief temporal window during development that ensures proper insulin secretion in adult life. Although adult pancreatic β-cells can undergo mitotic proliferation, less than 1% of adult pancreatic β-cells enter the cell cycle since as these cells become more functionally active, their proliferative capacity decreases [128]. However, periods of accelerated increase in β-cell mass can occur at times of intense metabolic demand, such as pregnancy, obesity, or insulin resistance (IR) [127].

Type 2 diabetes mellitus (T2DM) is a complex metabolic disorder characterized by increased IR in the liver, skeletal muscle, and adipose tissue, along with impaired insulin secretion by pancreatic β-cells, particularly in response to a glucose stimulus [129]. Type 2 diabetes mellitus is recognized as a progressive disease that is mainly related to a decrease in functional pancreatic β-cell mass over time [130]. Under an environment of IR, a compensatory mechanism is triggered to maintain glucose homeostasis by increasing insulin secretion. Thus, insulin exerts anti-apoptotic and proliferative effects to promote pancreatic β-cell hyperplasia. However, T2DM develops when the pancreatic β-cell mass becomes “exhausted” and cannot secrete adequate amounts of insulin to preserve normoglycemia. Type 2 diabetes mellitus is more common in patients with TS with a relative risk of 4.38 [131].

On the other hand, insulin, and insulin-like growth factor I and II (IGF-I and IGF-II) are members of the growth factor family and are implicated in the growth of virtually all tissues, especially fetal pancreatic β-cell development. They participate in several cellular mechanisms, such as antiapoptosis, protein synthesis, cell growth, and mitogenesis. Expression of *Igf1* in pancreatic β-cells from transgenic diabetic mice has been reported to regenerate the endocrine pancreas, likely by increasing pancreatic β-cell replication and neogenesis [132]. Although disturbances in the GH-IGF axis on pancreatic β-cell function in humans have been reported [128], a limited number of studies have explored the role of IGF-I in pancreatic β-cell function concerning glucose metabolism in TS. A negative correlation was found between plasma glucose and IGF-I in 30 subjects with TS studied by our group (unpublished data). Additionally, when IGF-I levels were adjusted for glucose concentration, a decrease in IGF-I was found in our adult TS subjects when compared with age- and BMI-matched healthy women. However, we did not find a relationship between IGF-I and insulin levels in our women with TS (unpublished data). A decrease in serum levels or the action of IGF-I could be detrimental to the already reduced cell replication in individuals with TS.

Under conditions of IR, pancreatic β-cells may compensate for the increase in insulin levels and sometimes their mass. However, in our study with women with TS, insulin levels failed to increase enough to maintain normal glucose levels during the Oral Glucose Tolerance Test (OGTT) [133]. This suggests deregulation of the pancreatic β-cell response to glucose overload and/or a decrease in the number of pancreatic β-cells. Although the antiapoptotic effect of IGF-I has been reported to be involved in the compensatory mechanism of pancreatic β-cells under conditions of IR or hyperglycemia, sustained plasma glucose has been associated with apoptosis leading to decreased β-cell mass [128]. Thus, four mechanisms may be involved in compensatory pancreatic β-cell failure in the face of glucose overload in adult subjects with TS: (a) a decrease in the β-cell mass due to a delay in the cell cycle during the fetal and early postnatal periods (Figure 5); (b) a reduction in the number of β-cells due to brakes on cell replication; (c) an increase in apoptosis; and (d) a decrease in IGF-I levels or its action.

Several processes combine to maintain an appropriate number of adult pancreatic β-cells, including decreased β-cell apoptosis and differentiation and dedifferentiation of progenitor cells from pancreatic ducts, acini, or even from bone marrow [132,134]. However, despite preexisting β-cell proliferation occurring at a low rate, this seems to be the main mechanism for the maintenance of adult β-cells [126,135]. Thus, the prolonged cell cycle hypothesis suggests that impaired cell proliferation is a critical process in the development of impaired glucose metabolism and T2DM in TS individuals, as it cannot ensure adequate cell proliferation during the fetal and early postnatal period and sufficient renewal in the adult life of pancreatic β-cells.

Similar to several studies, our data revealed increased basal insulin secretion in women with TS compared to age- and BMI-matched healthy women [133], which is a sign of the compensatory mechanism of pancreatic β-cells. However, our subjects with TS with impaired glucose tolerance and IR showed abnormalities in β-cell function during OGTT, suggesting that this compensatory mechanism could rapidly lead to “exhaustion” of pancreatic β-cells with subsequent development of hyperglycemia. Consequently, the increased compensatory mechanism of insulin-secreting β-cells to persistent IR could play a central role in the control of glucose homeostasis in subjects with TS. Therefore, decreased insulin secretion due to a lower pancreatic β-cell mass may be the most important determinant in the development of T2DM in TS.

### 3.6. Neurologic Deficits

#### 3.6.1. Neuropsychological Profile

Although it is difficult to assess which aspects may be associated directly with the genetic background, on average, TS individuals have a higher prevalence of developmental delays, including in the domains of language and fine and gross motor skills. As a group, girls with TS have mild but nonsignificant decreases in full-scale IQ when compared with age-matched peers [136]; thus, in neuropsychological studies of cognitive abilities, they are considered to have normal global intellectual functioning. Additionally, a particular pattern of cognitive strengths is repeatedly recognized in the verbal domains. However, nonverbal abilities are described to be significantly impaired [137]. Difficulties in higher-order visuospatial skills and arithmetical abilities are commonly described in neuropsychological studies carried out in girls with TS [138,139,140,141]. These difficulties may persist over time [141]. Spatial reasoning, mental rotations, visual attention, visual discrimination, visual sequencing, and visual-spatial memory are nonverbal neurocognitive skills identified as weaknesses in girls with TS compared to age-matched peers. Additionally, the deficit in executive functioning, social cognition, and an increased risk of autism spectrum conditions, attention-deficit/hyperactivity/impulsivity disorder, and potentially schizophrenia are reported in TS [136,141]. Although verbal abilities are typically normal, some elements of language development, such as verbal fluency, complex syntactic knowledge, and articulation, may be hindered by social and executive function deficits present in TS [141].

The neuropsychological profile reported in TS individuals is supported by neuroimaging studies. These clinical features probably reflect alterations in a complex network of brain areas, including the parietal lobe, prefrontal cortex, and amygdala [141,142]. Cerebrospinal fluid volume was reported to be increased by 25% in an affected prepubertal monozygotic twin compared with her sister with a corresponding decrease in gray matter volume [143]. A lower bilateral parietal volume, specifically in the superior parietal and postcentral gyri, was found in girls with TS than in age-matched peers [144,145]. In addition, a large-scale longitudinal study of brain volume growth over time in girls with TS reported aberrant neurodevelopment present early in childhood that extends into adolescence. Slower growth in the parieto-occipital gray and white matter areas during pubertal timing was observed in girls with TS than in girls with normal puberty [146]. Additionally, a general reduction in early visual areas in girls with TS was reported, which suggests that not only the parietal cortex, but also early visual areas are affected. This smaller cortical surface area may lead to less coverage of the peripheral visual field in subjects with TS compared to healthy controls [147]. Furthermore, a significantly smaller volume of the parietal-occipital brain matter was reported in TS women, which affects both gray and white matter [148].

On the other hand, several functional neuroimaging studies in subjects with TS have bilaterally described abnormalities in glucose metabolism and connectivity in the parietal and occipital cortex [145,148,149]. Glucose hypometabolism suggests impaired neuronal microstructural integrity (connectivity) of white matter pathways between the frontal and parieto-occipital regions [150,151]. Glucose hypometabolism is explained by a reduced neuronal density/mitochondrial function [145]. All these findings suggest that TS individuals may be susceptible to early and sustained impaired brain development in these cortical regions from birth to adulthood. These areas are associated with visuospatial cognition, processing, and reasoning. Similarly, the parietal lobe is involved in executive functioning and attention. Thus, these bilateral neuroimaging findings are consistent with impaired visuospatial and working memory skills and attention, which are repeatedly reported in TS individuals.

In summary, structural and functional neuroimaging studies have demonstrated a reduced parieto-occipital gray matter volume, impaired thickness and/or surface area of temporal-parieto-occipital cortical regions and decreased microstructural integrity of white matter. Although visual-spatial/perceptual dysfunction seems to be a hallmark of TS, the neuroanatomical and functional findings represent diffuse and multifocal brain abnormalities but not pathognomonic abnormalities of TS [152]. Therefore, this neuroimaging profile supports a genetic rather than an endocrine basis for these cognitive features seen in TS. The persistence of these anatomic abnormalities into adulthood reinforces this assumption and indicates that hormone replacement therapy does not have a major impact on the cognitive deficits in women with TS [153]. Thus, although neurocognitive deficits observed in TS could be caused by an early deficiency of sex steroids, they are present across wide age ranges and do not improve with estrogen treatment.

A genetic locus on the pseudoautosomal region 1 (PAR1) of the short arm of the X chromosome has been suggested to explain the TS cognitive profile [152]. However, the bilateral, diffuse, and multifocal cortical abnormalities aim to be a global rather than specific factor as to be causative of visual-spatial/perceptual deficits in TS. This was accompanied by a small, but proportional, decrease in gray matter volume. The findings mentioned above may also suggest that X monosomy leads to mild but widespread brain hypoplasia during neurodevelopment in subjects with TS, possibly due to specific features of cerebral cells during the critical window of perinatal development. The greater involvement of verbal domains than nonverbal domains may indicate the unavailability of optional processing routes in the temporal-parieto-occipital cortical regions rather than a different genetic constitution of brain cells. Although girls with TS with 45,X have been reported to be more severely affected than girls with TS with mosaicism [154], 45,X cells are unlikely to have a special predilection for these affected areas since the neuropsychological profile is consistently reported regardless of the genetic constitution of TS subjects. A complex interaction between genetic determinants and hormonal deficiencies has been proposed to explain the biological complexity of the neurocognitive developmental abnormalities in TS [142]. A critical region of <2 Mb within PAR1 has been associated with an increased risk of TS neurocognitive phenotype [152], but so far, no gene has been identified in this region that explains the visual-spatial/perceptual deficits seen in subjects with TS. Therefore, the etiology of the TS neuropsychological profile is most likely due to multiple genetic factors each contributing to the phenotypic variance. Hypothetically, the haploinsufficiency of loci on the Xp chromosome results in 0.5-fold diminished gene expression. However, transcriptome analyses in different human tissues and XO mouse models did not reveal a direct correlation between genomic imbalance and gene expression levels [56,57,58,59,60]. Thus, global cellular factors such as chromosome imbalance with impaired cell proliferation due to aneuploidy arise as an alternative hypothesis. Data from electrophysiological, neuroanatomical, and functional neuroimaging support this hypothesis. The smaller cortical surface area of early visual areas in girls with TS may be associated with a lower number of neurons, which, in turn, leads to less coverage of the peripheral visual field compared to controls [147]. Additionally, functional imaging studies in TS individuals suggest differences in neuronal membrane turnover and signal transduction that modify cell survival [145].

#### 3.6.2. Sensorineural Hearing Loss

Evidence that sensorineural hearing loss (SNHL) in TS is mediated by the effects of aneuploidy on rates of cell turnover was reported [155,156]. This hypothesis postulates that the greater the proportion of 45,X cells, the greater is the number of cells having both a prolonged cell cycle and a lack of transacting growth-regulating genes such as the SHOX gene. Thus, a prolonged cell cycle of 45,X cells leads to a smaller cochlea size and fewer auditory sensory cells (and their innervation), ganglion cells, and neurons at birth. Hence, in postnatal life, there is reduced differentiation and maturation of auditory ganglion cells [67]. Additionally, the increased prevalence of auricular anomalies in TS may be due to a failure in the upregulation of the cell cycle in the pharyngeal arches and the neck region. Since the density of hair cells is lower in the basal turn at birth than in the middle part of the organ of Corti, apoptosis due to age will be more deleterious to the basal than the middle part [157]. Consequently, hearing loss symptoms due to apoptosis of the mechanosensory hair cells induced by age or noise occur early in subjects with TS. On the other hand, the cell cycle delay hypothesis suggests that programming of the GH-IGF-1 axis induced by monosomy of the sex chromosome is an essential factor to explain the association between short stature and SNHL in TS and the general population [155,156]. Components of Metabolic Syndrome (MetS) are commonly associated with SNHL in the non-TS population. We reported a strong association between MetS and SNHL in adult subjects with TS [158]. Additionally, a significant relationship between height and SNHL was found in our study. Whether the risk factors for SNHL are confounding and not etiological mechanisms is an issue of discussion.

### 3.7. Cancer in Turner Syndrome under the Prolonged Cell Cycle Hypothesis

While the hypothesis of the delayed cell cycle in aneuploid cells assumes a slowing down of the cell division, and, therefore, to a reduction in the rate of cell proliferation, it is difficult to reconcile this hypothesis with the uncontrolled cell proliferation seen in neoplasms. No prospective studies of cancer occurrence in TS women have been published. Also, the risk of cancer in women with TS has been little explored, and the data published so far from retrospective observational studies are contradictory. A study with a national cohort of 3425 TS women established the overall risk of cancer was similar to that expected in the general population [159]. However, the overall risk of cancer is reported slightly raised in a retrospective observational study with standardized incidence ratios (SIR) between 0.9 and 1.34 [64]. This increased overall risk is principally due to site-specific cancers such as melanoma and meningioma. Remarkably, the incidence of breast cancer is repeatedly reported to be significantly reduced in TS women, a finding associated with low endogenous hypoestrogenism and the poor development of breast tissue seen in TS individuals. Thus, the cause of the increased risk of cancer in TS is unclear, and this may be due to be hormone-sensitive solid tumors.

The role of hormone-replacement therapy in the development of meningioma has been suggested [160,161]. Similarly, an increased but non-significant SIR for cutaneous melanoma has been noted [159]. Thus, the increased risk of melanoma is lesser than expected from the heightened number of pigmented nevi observed in women with TS [64]. Several studies have implicated a propensity to develop melanocytic nevi as an independent risk factor for cutaneous melanoma [162]. Melanocytes are pigment-producing cells derived from the neural crest and congenital nevi as those seen in TS probably represent an error in the development and migration of these neuroectodermal cells. The development of melanocytic nevi is multifactorial, heterogeneous, and the relationship among nevus evolution, anatomic location, and environmental and constitutional factors are complicated [163]. Growth factors have been suggested to be released by proliferating keratinocytes and could contribute to the stimulation of melanocyte proliferation.

Thus, other mutational factors must arise for the occurrence of neoplasms in TS individuals. These genetic alterations may override the low rates of cell growth and promote cell proliferation. If aneuploidy is present in somatic cells, it could result in genomic instability or apoptosis. The genomic instability involves both global hypomethylation and gene-specific hypermethylation, as well as widespread chromatin modifications. The genomic instability may lead to the gain of function mutations in one or more of several primary oncogenes. This oncogenic event could mark the transition to the next progression stage where secondary (or tertiary) oncogenic events may occur. These are commonly loss of function alterations of tumor suppressor genes. Remarkably, deletions in the X chromosome have been reported in meningioma, bladder cancer, and melanoma [164,165,166]. Therefore, other components beyond the delayed cell cycle appear to influence the development of tumor solid in TS. Either activation of an oncogene or the loss of a tumor suppressor gene would trigger an initial phase of proliferation for the establishment of a tumor solid.

## 4. Summary and Future Perspectives

Research on Turner syndrome, a condition affecting a substantial group of the female population throughout the world, has recently been focused on unconventional issues, such as modifications in gene expression, epigenetics, and the fetal programming hypothesis. The natural history of this disorder makes the analysis of this disease process challenging. Because the health consequences of TS diagnosis have been exhaustively recognized, it is essential to know the relationships between the pathogenesis of TS and overall health. The increasing number of studies and persistently refined findings indicate that the impacts of the available understanding are improving. The different phenotypic manifestations of TS seem to arise because of the interplay of diverse genetic factors that are usually first expressed in the embryonic preimplantation period. However, evidence for the prolonged cell cycle hypothesis in the pathogenesis of TS exists. There is a body of knowledge that shows that the TS phenotype may arise because of a prolonged cell cycle that could predispose individuals to the development of several clinical consequences both prenatally and postnatally (even in adult life). Given the heterogeneity seen in the findings of the studies discussed in this review, efforts to associate the prolonged cell cycle hypothesis with several clinical features are rarely uncomplicated. While we cannot definitively state that a prolonged cell cycle due to X chromosome monosomy causes different clinical features in TS, such as embryonic lethality, short stature, gonadal dysgenesis, osteopenia/osteoporosis, congenital heart diseases, neurologic deficits, sensorineural hearing loss, impaired pancreatic β-cell function, and other somatic problems, the studies included in this review demonstrate that the TS phenotype does not have an isolated or deterministic etiology but rather is part of a complex interaction of cellular events that may have an overall health impact on the affected individual. By continuing to assess specific subgroups of TS individuals, future studies may be able to better delineate the association between the prolonged cell cycle hypothesis and specific somatic features. Last, this literature review has several limitations. For example, it is possible that not all relevant studies were included, and publication and selection biases could prejudice the interpretation of the findings.

Several automated cell counting methods are commonly used to assess the cell proliferation rate, such as direct electrical impedance, flow cytometry, computer-aided image analysis, and serological counting, but, unfortunately, recent studies to evaluate the rate of cell proliferation in cells 45,X or with structural abnormalities of the sex chromosomes have not been reported. This is a limitation of the hypothesis suggested in this article. Thus, further studies will be needed to explain the proposed pathological mechanisms connecting the prolonged cell cycle and TS phenotype. For example, while the mechanisms underlying β-pancreatic cell loss in type 1 and 2 diabetes mellitus have been studied, less is explored about residual β-pancreatic cells. It is possible that β-pancreatic cells are “preserved” rather than irreversibly lost. Thus, studies that quantify the β-pancreatic cell proliferation rate in TS could lead to new approaches for potentially reactivating and preserving this cell mass.

## Figures and Tables

**Figure 1 jdb-10-00016-f001:**
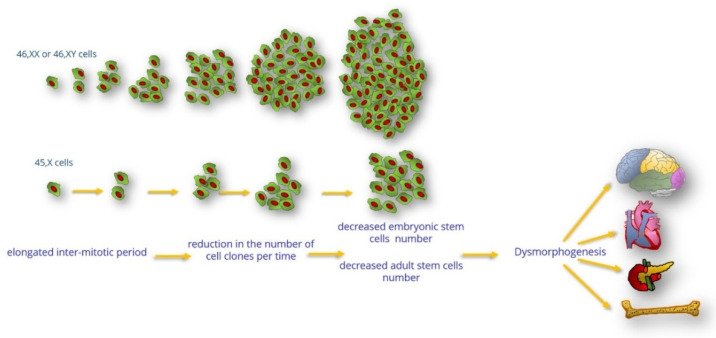
Consequences of the cell cycle duration between euploid and aneuploid cells. Cells with 45,X have a prolonged cell cycle, leading to a reduction in the number of cell clones per unit of time with selective disadvantages in vitro and in vivo.

**Figure 2 jdb-10-00016-f002:**
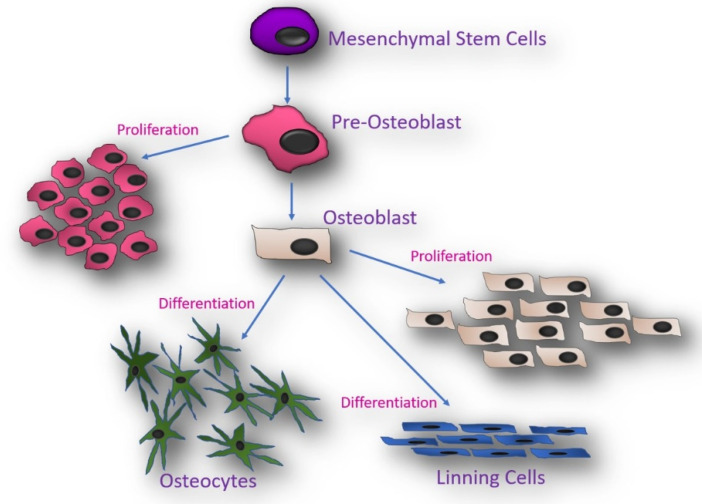
Development of bone precursor cells (mesenchymal stem cells, preosteoblasts, osteoblasts, mature osteocytes, and osteoclasts). This process is primarily mediated by osteoblasts. Both the proliferation and differentiation of osteoblast progenitors are essential for bone growth and development. Since the prolonged cell cycle of 45,X cells leads to fewer cell clones over time, the proliferating osteoblast progenitor population is affected, leading to a reduction in the number of bone-forming cells. Decreased proliferation of bone precursor cells might affect both bone mineralization and the skeleton (bone dysmorphogenesis and short stature).

**Figure 3 jdb-10-00016-f003:**
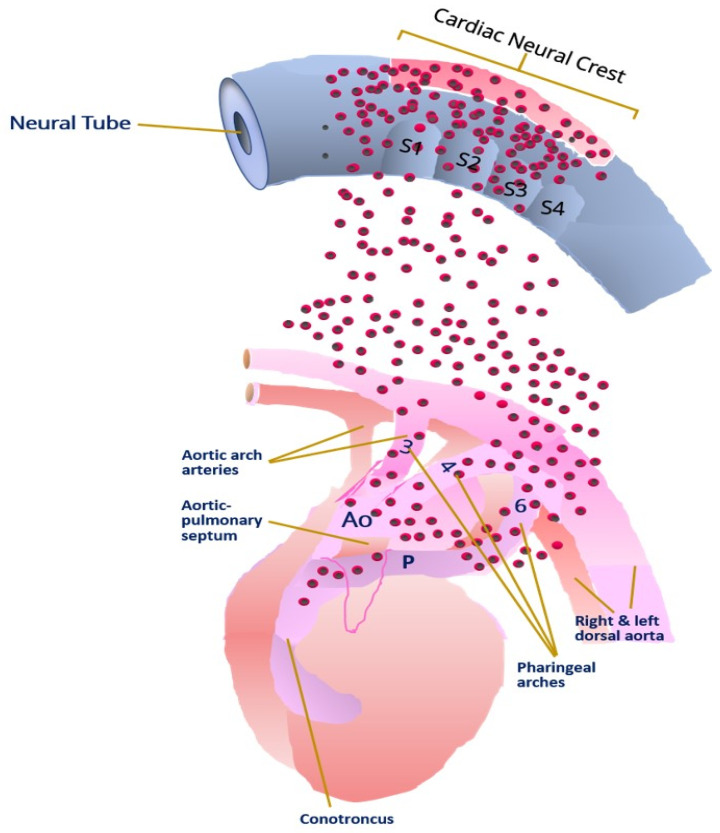
Graphic representation of the migration of the cardiac neural crest cells to the caudal pharyngeal arches and aorticopulmonary septum before remodeling of the aortic arch arteries and vascular regression. In 45,X embryos, when the delayed cell cycle results in a migratory reduction of cardiac neuronal cell proliferation, abnormal remodeling of the aortic arch arteries and/or vascular regression could originate as a subsequent consequence. Abbreviations: Ao, aorta; P, pulmonary artery; S1, S2, S3, and S4, somites 1–4; 3, 4, and 6, 3rd, 4th, and 6th pharyngeal arches.

**Figure 4 jdb-10-00016-f004:**
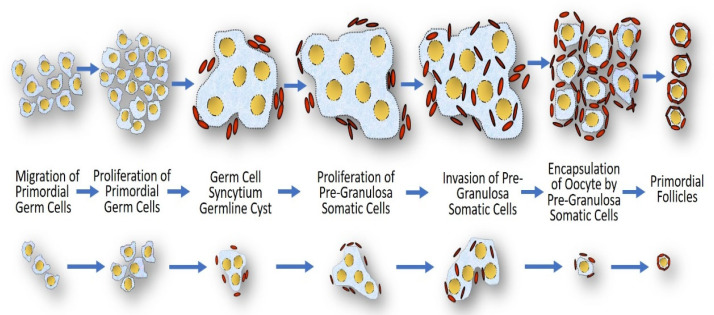
The process of development of the primordial follicle under the hypothesis of the prolonged cell cycle. After arrival at the gonads, the primordial germ cells form germline cysts. During this process, the pregranulosa cells proliferate by mitotic division to surround germ cells. Then, the first oogonia within the germline cysts enter meiosis and arrest at the end of prophase I. Later, the germline cysts are invaded by the pregranulosa cells to encapsulate oocytes and form primordial follicles. Both germ cells and pregranulosa cells need to enter successive cell divisions for proper assembly of primordial follicles to occur and to endow sufficient follicles for reproductive life. In Turner syndrome, a decrease in cell proliferation determines a decrease in the number of primordial cysts and inadequate assembly of the primordial follicles.

**Figure 5 jdb-10-00016-f005:**
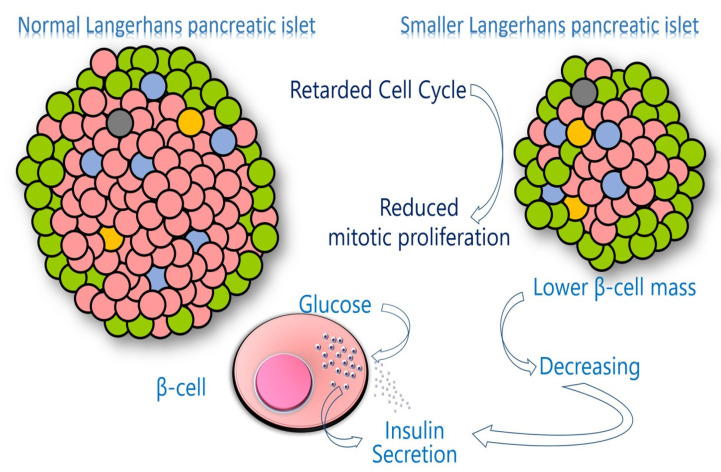
Development of type II diabetes mellitus in Turner syndrome. Located within the islets of Langerhans, pancreatic β-cells are sensitive to plasma glucose concentrations and secrete suitable amounts of insulin. Insulin exerts its action on the target peripheral tissues (liver, muscles, and adipocytes) to improve glucose utilization and storage. In Turner syndrome, a prolonged cell cycle leads to a decrease in and dysfunction of the pancreatic β-cell mass in an environment of increasing insulin resistance. Consequently, the lower pancreatic β-cell mass becomes “exhausted” early and cannot secrete adequate amounts of insulin to preserve normoglycemia.

## Data Availability

Not applicable.

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
