# Peer review of "The Hypothesis of the Prolonged Cell Cycle in Turner Syndrome"

_jdb, 2022, doi:10.3390/jdb10020016_

Round 1

Reviewer 1 Report

This revised review has significantly improved the quality of the manuscript. Please edit to correct typos, including in the references.

Reviewer 2 Report

Thanks to the authors, all my questions have been answered! They have put  a lot of work to the paper! The pictures are really nice, too!

This manuscript is a resubmission of an earlier submission. The following is a list of the peer review reports and author responses from that submission.

Round 1

Reviewer 1 Report

Thanks for the nice review on Turner syndrome!  I think that the value of the paper is in the clinical description of the many sides of TS. However, I did not find answers/ comments to some quite fundamental questions concerning the possible connection with missing one X- chromosome and cell cycle. In my mind, this hypothesis can well be put forward, but I find the review missing some key points to discuss. 

  1. How big is the difference in cell cycle between 45,X and 46,XX cells? Wouldn't even a small percentage difference  in the number of cells show a gross fetal size difference? What is the evidence of cell cycle differences in cell line measurements? The references are mainly quite old. It is quite acceptable to present hypotheses, but I'd be happy to learn what kind of cell cycle analyses and measurements have been done, because they are standard experiments in many laboratories. 
  2. What is known about X monosomy in animals?
  3. What is known about TS and cancer? If the cell cycle is slowlier, one would expect cancer incidence go down.
  4. One hypothesis is that 45,X cells are incapable to grow in early pregnancy. The hypothesis of a small (placental) marker chromosome rescuing the  1-1,5% is not mentioned. 
  5.  What is the role on X-inactivation? As lyonization happens in early blastocyst phase, what is the effect on 45,X cells during or after pregnancy? 

Reviewer 2 Report

This review has the potential to be useful to the field of Turner syndrome research as it does an excellent job of covering older literature that is focused on the prolonged cell cycle seen in studies of Turner syndrome cell lines done in the 1970s and 1980s.  I agree with the author that we should not ignore studies just because they were done decades ago.  And the "thought experiments" presented in this review linking cell cycle to different pathologies are not necessarily incorrect, but are deeply biased in favor of the author's pet hypothesis while almost completely neglecting a massive amount of recent literature on Turner syndrome.

The Turner syndrome research community would be better served by a more balanced article that focuses on reviewing the cell cycle literature, but also acknowledges the recent Turner syndrome literature and advances in the field. One suggestion is to look at the 2019 issue of the American Journal of Medical Genetics, Part C that is devoted to the coverage of the 2018 Turner Syndrome Resource Network conference proceedings. This would be a relatively easy way to come up to speed on the latest Turner syndrome research. I realize that this review article was a lot of work and was not intended to be a comprehensive review, but it should be possible to revise it to be more balanced.